# The Trans-Planckian Censorship Conjecture in Different Frameworks of Viable Inflation

Bruno Sanna [1] and Lorenzo Sebastiani [2,*]

1 Dipartimento di Fisica, Universitá di Pisa, Largo B. Pontecorvo 3, 56127 Pisa, Italy; b.sanna1@studenti.unipi.it
2 Istituto Nazionale di Fisica Nucleare, Sezione di Pisa, Largo B. Pontecorvo 3, 56127 Pisa, Italy
* Correspondence: lorenzo.sebastiani@pi.infn.it

**Abstract:** We review the recently proposed *Trans-Planckian Censorship Conjecture* (TCC) that stems from the trans-Planckian problem of cosmological perturbations. We analyze the implications and constraints that the TCC introduces in different frameworks of viable inflation. We revisit the case of slow-roll scalar field inflation and we investigate the cases of slow-roll $f(R)$ and $f(R, \phi)$-gravity. Finally, we consider the conjecture in the context of constant-roll scalar field inflation.

**Keywords:** trans-Planckian problem; inflation; constant-roll inflation

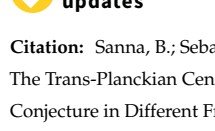



## 1. Introduction

The inflationary paradigm, according to which the Universe underwent a brief period of early-time rapid expansion, was initially introduced several years ago by Starobinsky [1,2], Guth [3] and Sato [4,5], and later by Linde [6], Albrech and Steinhardt [7]; in the last decades several theories have been suggested in order to describe inflation (see [8] for a nice review). Despite the fact that the arena of inflationary models is quite large, the huge amount of observational data [9,10] can be used to discriminate between the viable ones. In particular, inflation provides a causal mechanism to generate the primordial inhomogeneities across the matter distribution in our Universe, which evolve and persist in the Universe today and which are the object of cosmological observations.

Fluctuations in both matter and gravitational waves are believed to have a quantum mechanical origin in terms of vacuum perturbations that originate inside the Hubble radius (or horizon) at the beginning of inflation. During inflation, when they cross the Hubble horizon, they become classical and later re-enter the horizon [1,11]. The study of these perturbations can be carried out by making use of field-theory computations without invoking any trans-Planckian physics. However, inflationary cosmology generally suffers from the so called "trans-Planckian problem", which appears if the macroscopic fluctuations that cross the Hubble horizon trace back to trans-Planckian wavelengths at very early times. Since these fluctuations would contribute to the power spectrum, their computation involves low-energy physics into regions where this physics is not applicable, which is clearly not desirable [12–16], unless one admits the possibility that trans-Planckian effects manifest themselves in the form of ultra-high energy particles at any point in time [17,18]. Recently, Bedroya and Vafa have proposed an alternative viewpoint that avoids the trans-Planckian problem [19]. Their work is motivated by string theory, and is connected to the broader Swampland scenario, which encodes the low-energy effective field theories of gravity that are not compatible with (super)string theory [20,21]. In this respect, different Swampland conditions have been formulated during the years, such as the de Sitter Conjecture [22] and the Distance Conjecture [23], limiting the number of theories that admit an ultraviolet completion (or that belong to the "string landscape"). Some examples of constraints emerging from the Swampland criteria can be found in refs. [24,25].

The Trans-Planckian Censorship Conjecture (TCC) proposed by Bedroya and Vafa in the seminal paper [19] states that "*a field theory consistent with a quantum theory of gravity does not lead to a cosmological expansion where any perturbation with length scale greater than the Hubble radius traces back to trans-Planckian scales at an earlier time*". In other words, the TCC forbids Planck-scale perturbations to ever cross the Hubble horizon and enter the power spectrum. This statement can be formulated in the following mathematical form (in Planck units),

$$\frac{a(t)}{a(t_0)} l_{Pl} < \frac{1}{H(t)} \longrightarrow \frac{a(t)}{a(t_0)} < \frac{M_{Pl}}{H(t)}, \tag{1}$$

where $a(t)$ and $H \equiv H(t) = \dot{a}(t)/a(t)$ are the scale factor of the Universe and the Hubble parameter, respectively, at a generic time $t$, the dot being the derivative with respect to time, and $a(t_0)$ is the scale factor of the Universe at the early-time $t_0$, when quantum fluctuations take place. The Planck length $l_{Pl}$ is related to the Planck Mass $M_{Pl}$ as $l_{Pl} = 1/M_{Pl}$. As a consequence, when the TCC holds true, the length scales that exit the Hubble horizon preserve a wavelength bigger than the Planck length back into the past and the trans-Planckian quantum fluctuations remain quantum.

As an immediate consequence of the TCC we have that $H(t) < M_{Pl}$ in an expanding universe. Moreover, if the expansion is decelerated only, the TCC is never violated due to the fact that

$$\dot{a}(t) < \dot{a}(t_0) < a(t_0)M_{Pl} \longrightarrow a(t)H(t) < a(t_0)M_{Pl}. \tag{2}$$

Therefore, a possible violation of the TCC takes place if there is an accelerating expansion somewhere along the way.

As is well known, the expansion of our Universe today is accelerating and the so called "dark energy" epoch is well described by the Cosmological Constant, which implies a constant Hubble parameter (de Sitter space-time). Thus, if we assume the validity of the TCC we must accept that the de Sitter expansion cannot continue for an infinite amount of time and that there should be an upper bound for the lifetime of the Universe (in [19] it is estimated as $\sim 2.4$ trillion years). However, if the implications of the TCC on the late-time Universe are purely speculative, it is clear that they are extremely strong for inflation, which describes the early-time cosmic acceleration. In [26] it has been found that by assuming the TCC and in order to obtain a successful inflationary scenario for the structure formation of galaxies, the energy scale of inflation has to be lower than 109 GeV. Moreover, for slow-roll inflationary scalar field models, a negligible amplitude of primordial gravitational waves is predicted with a severe fine-tuning of initial conditions.

In this paper we would like to generalize these studies. We will investigate the impact of the TCC on different models and different frameworks of viable inflation. By "viable" we mean in agreement with cosmological observations, which constrain the values of the power spectrum of primordial fluctuations, the spectral index of scalar perturbations and the tensor-to-scalar power spectra ratio. Our aim is to analyze if, and under which conditions, inflation free of the trans-Planckian problem can be realized by starting from the TCC. In Section 2 we will revisit the consequences of the TCC in the classical picture of slow-roll scalar field inflation. In Sections 3 and 4 we will consider the cases of $f(R)$-gravity and $f(R, \phi)$-gravity, respectively. In Section 5 we will study the constant-roll scalar field inflationary scenario. Conclusions and final remarks are given in Section 6.

In our convention, the speed of light and the reduced Planck constant are $c = \hbar = 1$.

## 2. Scalar Field Slow-Roll Inflation

Let us start by considering a scalar field theory whose action is given by,

$$I = \int_{\mathcal{M}} d^4x \sqrt{-g} \left( \frac{M_{Pl}^2}{16\pi} R - \frac{g^{\mu\nu}\partial_\mu \phi \partial_\nu \phi}{2} - V(\phi) \right), \tag{3}$$

where $g$ is the determinant of the metric tensor and $V(\phi)$ is the potential of the scalar field $\phi$. The metric of a flat Friedmann–Robertson–Walker (FRW) space-time is given by,

$$ds^2 = -dt^2 + a(t)^2(dx^2 + dy^2 + dz^2).\qquad(4)$$

Thus, the first Friedmann equation is given by,

$$\frac{3H^2}{8\pi}M_{Pl}^2 = \frac{\dot{\phi}^2}{2} + V(\phi),\qquad(5)$$

with the associated field conservation law,

$$\ddot{\phi} + 3H\dot{\phi} = -\frac{dV(\phi)}{d\phi}.\qquad(6)$$

The dynamic of slow-roll inflation is described by the slow-roll parameters,

$$\epsilon_1 = -\frac{\dot{H}}{H^2}, \quad \epsilon_2 = \frac{\dot{\epsilon}_1}{H\epsilon_1},\qquad(7)$$

which should be small during inflation. Thus, the scale-invariant power spectrum of primordial fluctuations when their wavelength amplitudes are equal to the horizon size reads,

$$\mathcal{P} = \frac{1}{8\pi^2\epsilon_1}\left(\frac{H^2}{M_{Pl}^2}\right)_{k=aH},\qquad(8)$$

where $k$ is the wavenumber of perturbation.

Let us introduce the $e$-folds parameter,

$$N = \ln\left[\frac{a(t_e)}{a(t)}\right],\qquad(9)$$

where $t_e$ is the time when inflation ends. In order to solve the problem of the initial conditions of our Friedmann universe the perturbations must cross the horizon at $N \sim 55 - 65$ before the inflation ends. Thus, $N = 55 - 65$ is the minimum expansion rate required for viable inflation. According with the inhomogeneities observed in our Universe, $\mathcal{P} \sim 10^{-9}$, while the Planck data [9] constrain the spectral index of scalar perturbations $n_s$ and the tensor-to-scalar power spectra ratio $r$ as $n_s = 0.9649 \pm 0.0042$ at 68% CL and $r < 0.06$ at 95% CL. These quantities are given by (in first-order approximation),

$$n_s = 1 - 2\epsilon_1 - \epsilon_2, \quad r = 16\epsilon_1,\qquad(10)$$

and must be evaluated at $N = 55 - 65$.

Now we will see how the TCC Equation (1) introduces an upper bound for the Hubble parameter. We make use of the effective Equation of State (EoS) parameter,

$$\omega_{\text{eff}} = \frac{\dot{\phi}^2 - 2V(\phi)}{\dot{\phi}^2 + 2V(\phi)}.\qquad(11)$$

The slow-roll approximation $\epsilon_1, |\epsilon_2| \ll 1$ in Equations (5) and (6) leads to,

$$\frac{3H^2}{8\pi}M_{Pl}^2 \simeq V(\phi), \quad 3H\dot{\phi} \simeq -\frac{V(\phi)}{d\phi},\qquad(12)$$

such that

$$\epsilon_1 = \frac{3(1 + \omega_{\text{eff}}(N))}{2}, \quad \epsilon_2 = -\frac{d}{dN}\ln(1 + \omega_{\text{eff}}(N)),\qquad(13)$$

where we used the fact that $d/dt = -Hd/dN$ and made explicit the dependence of $\omega_{\text{eff}}$ on $N$. As a consequence we obtain

$$(1 - n_s) = 3(1 + \omega_{\text{eff}}(N)) - \frac{d}{dN} \ln(1 + \omega_{\text{eff}}(N)), \quad r = 24(1 + \omega_{\text{eff}}(N)), \quad (14)$$

with $N = 55 - 65$.

From Equation (8) together with Equation (1) we get,

$$\epsilon_1 \simeq \frac{10^9}{8\pi^2} \left( \frac{H^2}{M_{Pl}^2} \right) |_{k=aH} < \frac{10^9}{8\pi^2} e^{-2\mathcal{N}}, \quad (15)$$

where we assumed $\mathcal{P} \sim 10^{-9}$ and where $\epsilon_1$ must be evaluated at $N = 55 - 65$. Here, $\mathcal{N}$ is the total e-folds from the beginning of inflation and in order to satisfy the TCC we minimized the Hubble horizon $1/H(t)$ in Equation (1). Note that in any case the Hubble parameter should be almost a constant all through the inflation. Moreover, in order to check whether we meet the TCC condition, by taking into account that in slow-roll inflation the $\epsilon_1$ slow-roll parameter decreases with the *e*-folds number, we will take $\mathcal{N}$ as the e-folds when perturbations cross the horizon and we will pose $\mathcal{N} = 60$. Thus, we arrive at the following inequality,

$$1 > (12\pi^2)10^{-9}e^{2\mathcal{N}}(1 + \omega_{\text{eff}}(\mathcal{N})) \sim 10^{45}(1 + \omega_{\text{eff}}(\mathcal{N})), \quad (16)$$

which is our starting point to analyze viable scalar field inflation in terms of the effective EoS parameter. We will use a reconstructive approach following refs. [27,28].

Inflation corresponds to a (quasi) de Sitter space-time, when the effective EoS parameter can be taken close but not equal to the value of minus one. Since we need an exit from inflation we must also require $\omega_{\text{eff}} > -1$ (quintessence inflation), due to the fact that, if $\omega_{\text{eff}}$ passes through the value of minus one, the corresponding (exact) de Sitter space-time becomes a final attractor of the system and inflation never ends. Furthermore, we need $\omega_{\text{eff}}$ to approach $-1/3$ in order to eventually end acceleration. A reasonable ansatz for the EoS parameter in terms of the *e*-folds number is given by (see [27]),

$$\omega_{\text{eff}}(N) = \frac{\beta}{N^\alpha} - 1, \quad 0 < \alpha, \beta, \quad (17)$$

where $\alpha$ and $\beta$ are positive numbers. For large values of $N$ we have $\omega_{\text{eff}} \simeq -1$, while acceleration ends when $N \to 0$. As a consequence, the spectral index and the tensor-to-scalar spectra ratio (14) are derived as,

$$(1 - n_s) = \frac{3\beta}{\mathcal{N}^\alpha} + \frac{\alpha}{\mathcal{N}}, \quad r = \frac{24\beta}{\mathcal{N}^\alpha}. \quad (18)$$

Since we are considering $\mathcal{N} = 60$, the spectral index $n_s$ satisfies the Planck constraint only if $\alpha = 1$ or $\alpha = 2$, but in the first case $\beta$ should be $\beta \simeq 1/3$ and the tensor-to-scalar ratio is ruled out by observations. A scalar field equation with an effective EoS parameter in the form of Equation (17) with $\alpha = 1$ corresponds to power-law potentials [28] and the choice $\beta = 1/3$ leads to a quadratic potential, whose viability fell down due to the incompatibility with the observed tensor-to-scalar spectra ratio.

Thus, we will focus on the case $\alpha = 2$, namely

$$\omega_{\text{eff}}(N) = \frac{\beta}{N^2} - 1, \quad \beta > 0, \quad (19)$$

in order to have

$$(1 - n_s) \simeq \frac{2}{\mathcal{N}}, \quad r = \frac{24\beta}{\mathcal{N}^2}, \quad (20)$$

which are in general in agreement with the Planck data.

The TCC condition of Equation (16) reads,

$$\beta^{-1} > \frac{10^{45}}{\mathcal{N}^2} \simeq 3 \times 10^{41} , \tag{21}$$

and we find the following upper bound on the parameter $\beta$,

$$\beta < 3 \times 10^{-42} . \tag{22}$$

This results in

$$r < 10^{-44} , \tag{23}$$

confirming the severe fine-tuning of initial conditions found in [26]. However, we can explicitly reconstruct a viable model that is compatible with the TCC predicting a strong suppression of the amplitude of primordial gravitational waves. As a matter of fact, the EoS parameter in Equation (19) corresponds to an exponential potential, as we can easily verify. By using the prime index to denote the derivative with respect to the *e*-folds number, in the slow-roll approximation of Equation (12) we derive,

$$(1 + \omega_{\text{eff}}(N)) \simeq 2H'/(3H) , \tag{24}$$

such that by using Equation (19) we obtain a differential equation for $H$ whose solution reads

$$H^2 \simeq \frac{8\pi\rho_0}{3M_{Pl}^2} e^{-\frac{3\beta}{N}} . \tag{25}$$

Here, $\rho_0$ is an integration constant whose physical meaning is the effective energy density of the Universe at the beginning of inflation, when $\mathcal{N}$ is quite large. Now, by equaling $3H^2 M_{Pl}^2/(8\pi)$ to $V(\phi)$ we obtain, in slow-roll approximation,

$$\omega_{\text{eff}} \simeq -1 + \frac{1}{9\beta} \ln^2 \left[ \frac{V(\phi)}{\rho_0} \right] , \tag{26}$$

and together with Equation (11) we get

$$\dot{\phi} \simeq -\frac{\sqrt{V(\phi)}}{3\sqrt{\beta}} \ln \left[ \frac{V(\phi)}{\rho_0} \right] \simeq \frac{\sqrt{V(\phi)}}{3\sqrt{\beta}} \left[ \frac{\rho_0}{V(\phi)} - 1 \right] , \tag{27}$$

where we assume $\dot{\phi} > 0$ during inflation, when $V(\phi)$ is smaller and close to the initial (effective) energy density $\rho_0$. Now, by making use of the second equation in Equation (12) we are able to reconstruct the full form of the scalar field potential as

$$V(\phi) = \rho_0 \left( 1 - c_1 e^{\sqrt{\frac{8\pi}{3\beta}} \frac{\phi}{M_{Pl}}} \right) , \tag{28}$$

where $c_1$ is a positive dimensional integration constant and we are taking $\phi < 0$.

In terms of the cosmological time, the explicit solutions $H \equiv H(t)$ and $\phi \equiv \phi(t)$ of Equation (12) are given by,

$$H(t)^2 = \frac{8\pi\rho_0}{3M_{Pl}^2} \left( 1 - \frac{3\sqrt{3}\beta M_{Pl}}{\sqrt{8\pi\rho_0}(t_e - t)} \right) , \quad \phi(t) = -\sqrt{\frac{3\beta}{8\pi}} M_{Pl} \ln \left[ \frac{c_1 \sqrt{8\pi\rho_0}}{3\sqrt{3}\beta M_{Pl}} (t_e - t) \right] , \tag{29}$$

where $t_e$ is approximately the time when inflation ends and $\beta M_{Pl}/(c_1\sqrt{\rho_0}) \ll t_e$ such that $|\phi|/M_{Pl} \gg 1$ at the beginning of inflation. The *e*-folds $N \equiv N(t)$ is given by,

$$N(t) \simeq \frac{8\pi}{M_{Pl}^2} \int_{\phi(t_e)}^{\phi(t)} \frac{V(\phi)}{V'(\phi)} d\phi \simeq \frac{3\beta}{c_1} e^{-\sqrt{\frac{8\pi}{3\beta}} \frac{\phi(t)}{M_{Pl}}} = \sqrt{\frac{8\pi\rho_0}{3}} \frac{(t_e - t)}{M_{Pl}} , \tag{30}$$

and the total amount of inflation is

$$\mathcal{N} \equiv N(0) \simeq \sqrt{\frac{8\pi\rho_0}{3}} \frac{t_e}{M_{Pl}} . \tag{31}$$

Finally, the slow-roll parameters of Equation (7) in terms of the cosmological time read

$$\epsilon_1 = \frac{9\beta M_{Pl}^2}{16\pi\rho_0(t_e - t)^2} , \quad \epsilon_2 = \frac{2}{(t_e - t)} \sqrt{\frac{3M_{Pl}^2}{8\pi\rho_0}} . \tag{32}$$

As we observed above, when $\mathcal{N} = 60$, the TCC condition is satisfied for $\beta < 3 \times 10^{-42}$. This means that the Hubble parameter is a constant during almost all the early-time acceleration and only at the very end of inflation goes to zero.

As a last remark we note that the model with $c_1 = 2$ and $\beta = 1/2$ corresponds to a scalar field inflation of the Starobinsky model in the Einstein frame [28], which clearly violates the TCC condition. In the next sections, we will investigate the TCC in inflationary modified gravity theories frameworks.

### 3. The Case of $f(R)$-Gravity

A different approach to inflation is given by the modified theories of gravity, where the gravitational Lagrangian is described as a general function of some curvature invariants. Generally speaking, one expects that at the early time some corrections to Hilbert–Einstein action arise, maybe related to quantum effects at high curvature [29–31]. In this Section we would like to analyze the simplest class of such models, namely $f(R)$-gravity, where the Lagrangian depends on the Ricci scalar only [32–37].

Let us consider the gravitational action,

$$I = \frac{M_{Pl}^2}{16\pi} \int_{\mathcal{M}} d^4 \sqrt{-g} \, f(R) , \tag{33}$$

where $f(R)$ is a function of the Ricci scalar $R$. The first Friedmann-like equation is given by,

$$3FH^2 = \frac{(FR - f)}{2} - 3H\dot{F} , \tag{34}$$

with $F = df/dR$, $f \equiv f(R)$.

In the framework of $f(R)$-gravity, slow-roll inflation is described by the slow-roll parameters [38–40],

$$\epsilon_1 = -\frac{\dot{H}}{H^2} \simeq -\epsilon_3(1 - \epsilon_4), \quad \epsilon_4 = -3\epsilon_1 + \frac{\dot{\epsilon}_1}{H\epsilon_1} , \tag{35}$$

whose magnitude is assumed to be small during inflation[1]. We note that in the first-order approximation the $\epsilon_3$ slow-roll parameter coincides with the opposite value of the $\epsilon_1$ slow-roll parameter and in the following expressions for the power spectrum and the spectral index we will pose $\epsilon_3 \simeq -\epsilon_1 \simeq \dot{H}/H^2$. However, the tensor-to-scalar spectra ratio must be evaluated at the second leading order of $\epsilon_1 + \epsilon_3 \simeq (\dot{H}/H^2)\epsilon_4$, which implicitly defines $\epsilon_3$.

The power spectrum of cosmological perturbations is given by [32]

$$\mathcal{P} = \frac{1}{24\pi^2 F\epsilon_1^2} \left(\frac{H}{M_{Pl}}\right)^2_{k=aH} \simeq \frac{1}{24\pi^2 F\epsilon_3^2} \left(\frac{H}{M_{Pl}}\right)^2_{k=aH} , \tag{36}$$

---

[1]  In the next section we will consider the more general framework of $f(R, \phi)$-gravity, which includes $f(R)$-gravity as a special case. Thus, $\epsilon_1, \epsilon_3$ and $\epsilon_4$ are labeled according to the corresponding slow-roll parameters in $f(R, \phi)$-gravity.

while the spectral index and the tensor-to-scalar power spectra scalar ratio read (in the first- and second-order approximations),

$$n_s = 1 - 4\epsilon_1 + 2\epsilon_3 - 2\epsilon_4 \simeq 1 - 6\epsilon_1 - 2\epsilon_4, \quad r = 16(\epsilon_1 + \epsilon_3) \simeq 48\epsilon_1^2. \tag{37}$$

As well as in the previous case, it is convenient to introduce an effective equation of state parameter as in Equation (24). In terms of the *e*-folds we get,

$$1 - n_s = -2\frac{d}{dN}\ln(1 + \omega_{\text{eff}}(N)), \quad r = 108(1 + \omega_{\text{eff}}(N))^2, \tag{38}$$

with $N = 55 - 65$. Thus, the TCC condition holds true if

$$1 > 54\pi^2 10^{-9} e^{2\mathcal{N}} F(\mathcal{N})(1 + \omega_{\text{eff}}(\mathcal{N}))^2 \sim 6 \times 10^{45} F(\mathcal{N})(1 + \omega_{\text{eff}}(\mathcal{N}))^2. \tag{39}$$

Since we are interested in the sufficient condition to meet the TCC condition, we again pose the total *e*-folds from the beginning of inflation equal to the *e*-folds when perturbations cross the horizon, namely $\mathcal{N} = 60$, and we consider the implicit form of $F$ as a function of $\mathcal{N}$.

As in Section 2, we can assume the ansatz of Equation (17) for the effective EoS parameter $\omega_{\text{eff}}(N)$. As a consequence we obtain

$$(1 - n_s) = \frac{2\alpha}{\mathcal{N}}, \quad r = \frac{108\beta^2}{\mathcal{N}^{2\alpha}}. \tag{40}$$

In this case the Planck constraint on $n_s$ implies $\alpha = 1$ [28,35],

$$\omega_{\text{eff}}(N) = \frac{\beta}{N} - 1, \quad \beta > 0, \tag{41}$$

which leads to

$$(1 - n_s) = \frac{2}{\mathcal{N}}, \quad r = \frac{108\beta^2}{\mathcal{N}^2}. \tag{42}$$

This choice corresponds to the Hubble parameter,

$$H^2 = \frac{8\pi}{3M_{Pl}^2}\rho_e N^{3\beta}, \tag{43}$$

which follows from Equation (24). Here, $\rho_e$ is an integration constant representing the effective energy density of the Universe at the end of inflation. Now we can infer the implicit form of $F(N)$ from Equation (34), which reads

$$-4HH'(F - 1) + 2H^2 F' + 2H^2 F'' + 2HH'F' = \frac{16\pi\rho_e\beta}{M_{Pl}^2}, \tag{44}$$

with $F \equiv F(N)$. A simple analytic solution can be found for $\beta = 1/3$, namely

$$F = c_0\left(\frac{1}{2} + N\right). \tag{45}$$

By taking into account that

$$R = 12H^2 + 6\dot{H} = 12H^2 - 6HH', \tag{46}$$

we easily derive

$$F = \frac{c_0}{2} + \frac{3M_{Pl}^2 c_0}{18(8\pi)\rho_e}R. \tag{47}$$

The $f(R)$-model can finally be fully reconstructed as

$$f = R + \frac{M_{Pl}^2}{48\pi\rho_e}R^2,$$

(48)

where we posed $c_0 = 2$ in order to recover the Hilbert–Einstein term of General Relativity (GR). This model is nothing else but the Starobinski model [2], which clearly violates the TCC condition of Equation (39). This fact is not surprising, since the Starobinsky model in the Einstein-frame leads to the scalar model with potential in Equation (28) and $\beta = 1/2$, $c_1 = 2$. The conformal transformation between the two frames is given by,

$$\phi(N) = -\sqrt{\frac{3M_{Pl}^2}{16\pi}} \ln F(N).$$

(49)

Now it is easy to verify that the power spectra of the two models coincide after the identification $\rho_e = 3\rho_0/2$ (note that the inflation scales in the two frames do not coincide).

One may be interested to see if Equation (44) admits some solutions for small values of $\beta$, when the Hubble parameter reads

$$H \simeq \sqrt{\frac{8\pi\rho_e}{3M_{Pl}^2}} \left(1 + \frac{3\beta}{2}\log N\right).$$

(50)

Thus, an implicit solution of Equation (44) to the leading-order term of $\beta$ that allows recovery of the Hilbert–Einstein contribution of GR is given by,

$$F = 1 + 3\beta N.$$

(51)

In this case the TCC condition of Equation (39) is satisfied under the condition

$$\beta < 8 \times 10^{-22},$$

(52)

which brings to

$$r < 2 \times 10^{-44}.$$

(53)

Due to the constraint on $\beta$ the Hubble parameter remains a constant during inflation and starts to decrease at the very end of it. Moreover, as in the case of scalar field slow-roll inflation, in this class of viable $f(R)$-gravity models compatible with the TCC we have a strong suppression of the amplitude of the primordial gravitational waves and a severe fine-tuning of initial conditions.

Despite the fact that our analysis is not exhaustive of the wide variety of $f(R)$-models for inflation, we can draw some conclusions. Since viable $f(R)$-gravity reduces to Einstein gravity at small curvature, it is clear that the TCC condition given in Equation (39) introduces an upper bound on the effective EoS parameter as $(1 + \omega_{\text{eff}}) \lesssim 10^{-23}$, such that the tensor-to-scalar spectra ratio in Equation (37) is at most $r \sim 10^{-44}$. This result is independent of the ansatz on $\omega_{\text{eff}}$ and is in agreement with Equations (52) and (53). Thus, the suppression of the amplitude of the primordial gravitational waves seems to be a general feature of viable slow-roll $f(R)$-gravity compatible with TCC and since we have assumed as a minimal requirement that the total amount of inflation coincides with the $e$-folds at the perturbation horizon crossing, we get the fine-tuning problem of initial conditions.

## 4. The Case of $F(R, \phi)$-Gravity: Two Specific Examples of Slow-Roll Inflation

An important class of inflationary models is given by scalar–tensor theories, where the gravitational interaction is mediated by both a scalar and a tensor field [41,42]. In what follows, in our attempt to investigate the TCC in this framework, we will consider two

specific examples of $f(R, \phi)$-slow-roll inflation firstly presented in [43], where a scalar field is coupled with the Ricci scalar.

The general action of $F(R, \phi)$-gravity is in the form,

$$I = \int_{\mathcal{M}} dx^4 \sqrt{-g} \left[ \frac{M_{Pl}^2 f(R, \phi)}{16\pi} - \frac{g^{\mu\nu} \partial_\mu \phi \partial_\nu \phi}{2} - V(\phi) \right], \tag{54}$$

but in what follows we will assume $V(\phi) = 0$.

In terms of the *e*-folds number, the slow-roll parameters describing slow-roll $f(R, \phi)$-inflation are given by [38],

$$\epsilon_1 = \frac{H'}{H}, \quad \epsilon_2 = -\frac{HH'\phi' + H^2\phi''}{H^2\phi'}, \quad \epsilon_3 = -\frac{F'}{2F}, \quad \epsilon_4 = -\frac{E'}{2E}, \tag{55}$$

where $F = \partial f / \partial R$, $f \equiv f(R, \phi)$, and

$$E = F + \frac{3M_{Pl}^2 F'^2}{16\pi \phi'^2}. \tag{56}$$

The first Friedmann-like equation in a slow-roll approximation leads to

$$3FH^2 \simeq \frac{1}{2}(RF - f), \tag{57}$$

while the conservation law related to the field reads,

$$-3H^2\phi' \simeq \frac{1}{2} \frac{\partial f}{\partial \phi}. \tag{58}$$

The power spectrum of cosmological perturbations is given by,

$$\mathcal{P} = \frac{1}{32\pi^3 Q_s} H^2 |_{k=aH}, \tag{59}$$

with

$$Q_s = \frac{\phi'^2 E}{F}. \tag{60}$$

As a check, we observe that when $f(R, \phi) = R$ and therefore $Q_s = \phi'^2$, in a slow-roll approximation we recover Equation (8). On the other hand, if $f(R, \phi) = f(R)$, $\phi' = 0$ and therefore $Q_s = 3M_{Pl}^2 F'^2/(16\pi F)$, we recover Equation (36).

The TCC introduces the following constraint,

$$Q_s < \frac{10^9}{32\pi^3} M_{Pl}^2 e^{-2\mathcal{N}}, \tag{61}$$

with $\mathcal{N} = 60$, as per usual. Finally, we recall the expressions for the spectral index and the tensor-to-scalar power spectra ratio,

$$n_s = 1 - 4\epsilon_1 - 2\epsilon_2 + 2\epsilon_3 - 2\epsilon_4, \quad r = 16(\epsilon_1 + \epsilon_3), \tag{62}$$

with $N = 55 - 65$. As a check, we note that when $f(R, \phi) = R$ such that $\epsilon_3 = \epsilon_4 = 0$, by taking into account that, in slow-roll approximation, $2\epsilon_2 = -\epsilon_1 - H''/H'$, we correctly find the results of slow-roll scalar field inflation in Equation (10) where $\epsilon_2 = H'/H - H''/H$. Moreover, if we pose $f(R, \phi) = f(R)$, $\phi' = 0$, we get [35] $\epsilon_2 = 0$, $\epsilon_1 \simeq -\epsilon_3(1 - \epsilon_4)$ and, by taking into account that in slow-roll approximation $\epsilon_1 \simeq -\epsilon_3$ and $\epsilon_4 \simeq -3\epsilon_1 - \epsilon_1'/\epsilon_1$, we recover the results of pure $f(R)$-gravity in Equation (37).

In [43] inflation is realized thanks to a sort of switching on the cosmological constant in two different models. The first model reads,

$$f(R,\phi) = R - 2\lambda \left( 1 - e^{b\left(\frac{8\pi}{3M_{Pl}^2}\right)^3 \phi R} \right), \quad b > 0, \tag{63}$$

where $b$ is a positive parameter and $\lambda$ is a positive cosmological constant (on the curvature scale of inflation). Inflation starts with $\phi$ negative and very small ($|\phi| \ll 0$), such that we obtain a quasi-de Sitter expansion with

$$H^2 \simeq \frac{\lambda}{3}. \tag{64}$$

Thus, from Equation (58) with $R \simeq 12H^2$ we obtain,

$$\phi = -\frac{\log\left[16b^2\left(\frac{8\pi}{3M_{Pl}^2}\right)^6 \lambda^2 N\right]}{4b\left(\frac{8\pi}{3M_{Pl}^2}\right)^3 \lambda}. \tag{65}$$

The field grows during inflation, which ends at $N \to 0$. A direct evaluation of the slow roll parameters shows that

$$\epsilon_1, \epsilon_3, \epsilon_4 \sim \frac{M_{Pl}^4}{b^2\lambda^2 N^2}, \quad \epsilon_2 = \frac{1}{N}. \tag{66}$$

Thus, a viable scenario with

$$1 - n_s \simeq \frac{2}{\mathcal{N}}, \quad r \sim \frac{M_{Pl}^4}{b^2\lambda^2\mathcal{N}^2}, \tag{67}$$

takes place. Moreover, since $Q_s \simeq \phi'^2$, we observe that the TCC condition of Equation (61) holds true if

$$\frac{M_{Pl}^4}{b^2\lambda^2\mathcal{N}^2} < 256 \times 10^9 e^{-2\mathcal{N}} \simeq 2 \times 10^{-41}, \tag{68}$$

and the amplitude of primordial gravitational waves is again strongly suppressed. By taking into account that the $R \sim \lambda$ condition of Equation (68) also guarantees that the Hubble parameter is almost a constant until the very end of inflation, due to the fact that $f$ turns out to behave as $f \simeq R - 2\lambda$, unless $N$ is very close to zero.

The second model under consideration reads,

$$f(R,\phi) = R - 2\lambda \left( 1 - \frac{1}{1 + \left(-b\left(\frac{8\pi}{3M_{Pl}^2}\right)^3 \phi R\right)^n} \right), \quad n > 0, b > 0, \tag{69}$$

with $n, b$ positive parameters and $\lambda$ a cosmological constant. Once again, inflation is supported by a quasi-de Sitter solution with $\phi$ negative and very small such that

$$H^2 \simeq \frac{\lambda}{3}. \tag{70}$$

The field behaves as,

$$\phi = -\frac{(2+n)^{\frac{1}{2+n}}(nN)^{\frac{1}{2+n}}}{\left(4\lambda b\left(\frac{8\pi}{3M_{Pl}^2}\right)^3\right)^{\frac{n}{2+n}}}, \tag{71}$$

and the early-time acceleration ends when $N \to 0$. The slow-roll parameters are derived as

$$\epsilon_1 , \epsilon_3 , \epsilon_4 \sim \frac{M_{Pl}^4}{b^2 \lambda^2 N^2}, \quad \epsilon_2 = \frac{1}{N} \left( \frac{1+n}{2+n} \right). \tag{72}$$

As a consequence,

$$1 - n_s \simeq \frac{2}{N} \left( \frac{1+n}{2+n} \right) \simeq \quad r \sim \frac{M_{Pl}^4}{b^2 \lambda^2 N^2}, \tag{73}$$

and the spectral index $n_s$ is in agreement with the Planck observations only for large values of $n$. In this case the TCC is satisfied under the requirement

$$\frac{M_{Pl}^4}{(n+2)^2 b^2 \lambda^2 (\mathcal{N})^2} < 8^3 \times 10^9 e^{-2\mathcal{N}} \simeq 4 \times 10^{-41}, \tag{74}$$

confirming the suppression of the amplitude of primordial gravitational waves as the price to pay for the validity of the TCC condition.

Up to now we have considered models of slow-roll inflation. The slow-roll approximation is valid if all the slow-roll parameters are small during inflation. However, in order to obtain a constant Hubble parameter (or a flat potential, in the classical scenario of scalar field inflation) it is enough to require that $\epsilon_1 \ll 1$, while the other horizon flow parameters can also be not so small, but constant. In the next Section, we will study the consequences of the TCC in the case of the so called "constant-roll" inflation scenario.

## 5. Constant-Roll Scalar Field Inflation

Constant-roll inflation has some important and interesting properties. For example, it can generate large local non-Gaussianities (which are negligible in the case of slow-roll inflation) and the curvature perturbations may grow on super-horizon scales [44–47]. In [48] constant-roll scalar field inflation has been studied and exact solutions for the inflaton potential have been found (see also refs. [49–52] for constant-roll inflation in modified gravity). We recall these results in the context of the TCC.

The action of the theory is still given by Equation (3). Scalar field constant-roll inflation takes place when $\ddot{\phi} \sim H\dot{\phi}$, being non-negligible in Equation (6). Following the prescription used in [48] we assume

$$\ddot{\phi} = -(3 + \alpha) H \dot{\phi}. \tag{75}$$

For $\alpha = -3$ we obtain the slow-roll approximation. We will investigate two models, for which all the possible values of $\alpha \neq -3$ are covered. The first model reads,

$$V(\phi) = 3M^2 M_{Pl}^2 \left[ 1 + \frac{\alpha}{6} \left( 1 - \cosh \sqrt{2(3+\alpha)} \frac{\phi}{M_{Pl}} \right) \right], \quad -3 < \alpha, \tag{76}$$

where $0 < M$ is a generic mass constant. We assume $-\infty < \phi < 0$ and $0 < \dot{\phi}$ during inflation. If $-3 < \alpha < 0$ the potential has a minimum (i.e., an attractor point) for $\phi \to 0^-$, while if $0 < \alpha$ the field reaches a maximum of the potential when $\phi \to 0^-$.

The second model is described by the following field potential,

$$V(\phi) = 3M^2 M_{Pl}^2 \left[ 1 + \frac{\alpha}{6} \left( 1 - \cos \sqrt{-2(3+\alpha)} \frac{\phi}{M_{Pl}} \right) \right], \quad \alpha < -3. \tag{77}$$

Here, we are assuming $\infty > \phi > 0$, $\dot{\phi} > 0$ and the potential has a maximum at $\phi = 0^+$. The exact solutions of the field Equations (5) and (6) in the case of Equation (76) are

$$
\begin{aligned}
H(t)^2 &= M^2 \coth^2[(3+\alpha)Mt], \\
\phi(t) &= M_{Pl} \sqrt{\frac{2}{3+\alpha}} \ln \left[ \coth \left[ \frac{(3+\alpha)}{2} Mt \right] \right],
\end{aligned}
\tag{78}
$$

after rescaling of $M^2 \to M^2/(8\pi)$. When $t \to 0^+$ the Hubble parameter is not a constant (but we can still have an acceleration). Nevertheless, since the Hubble parameter approaches a constant when $t \to +\infty$ and the scale factor grows exponentially, we eventually have inflation. Additionally, in this case a transition phase at the end of inflation has to be assumed (see refs. [53–55]).

The exact solutions of Equations (5) and (6) in the case of Equation (77) are

$$
\begin{aligned}
H(t)^2 &= M^2 \tanh^2[(3+\alpha)Mt], \\
\phi(t) &= 2M_{Pl}\sqrt{-\frac{2}{3+\alpha}} \arctan\left[e^{-(3+\alpha)Mt}\right],
\end{aligned}
\tag{79}
$$

with $M^2 \to M^2/(8\pi)$ again and $-\infty < t < 0$, such that the Hubble parameter is almost a constant when $t \to -\infty$ and we have inflation, while it goes to zero when $t \to 0^-$.

We will denote with $t_0$, $t_e$ the time when inflation starts and ends, respectively. The $e$-folds number of Equation (9) reads,

$$
N = \int_t^{t_e} H dt.
\tag{80}
$$

For the model in Equation (76) we obtain,

$$
N = \frac{1}{(3+\alpha)} \log\left[\frac{(a(t_e)/a(t_0))^{3+\alpha}}{\sinh[(3+\alpha)Mt]}\right] \to t = \frac{\text{arcsinh}\left[e^{-(3+\alpha)N}(a(t_e)/a(t_0))^{3+\alpha}\right]}{M(3+\alpha)},
\tag{81}
$$

where we have evaluated $a(t) = a(t_0)(\sinh[(3+\alpha)Mt])^{1/(3+\alpha)}$ and we have posed $\sinh[(3+\alpha)Mt_0] = 1$.

For the model in Equation (77) we derive,

$$
N = \frac{1}{(3+\alpha)} \log\left[\frac{1}{\cosh[(3+\alpha)Mt]}\right] \to t = \frac{\text{arccosh}\left[e^{-(3+\alpha)N}\right]}{M(3+\alpha)},
\tag{82}
$$

where we have evaluated $a(t) = a(t_e)(\cosh[(3+\alpha)Mt])^{1/(3+\alpha)}$ and we have posed $\cosh[(3+\alpha)Mt_e] = 1$ (namely $t_e = 0$).

We can now estimate the $\epsilon_1$ slow-roll parameter of Equation (7) in the two models as

$$
\epsilon_1 = \frac{3+\alpha}{\cosh^2[(3+\alpha)Mt]} = \frac{3+\alpha}{1+e^{2(3+\alpha)(\mathcal{N}-N)}}, \quad -3 < \alpha,
\tag{83}
$$

$$
\epsilon_1 = -\frac{3+\alpha}{\sinh^2[(3+\alpha)Mt]} = \frac{3+\alpha}{1-e^{-2(3+\alpha)N}}, \quad \alpha < -3,
\tag{84}
$$

where we have introduced the total $e$-folds number $\mathcal{N}$ through the relation in Equation (9). Thus, for $-3 < \alpha$, the bound of the $\epsilon_1$ slow-roll parameter at the time $t = t_0$ (namely, $N = \mathcal{N}$) is given by $\epsilon_1 = (3+\alpha)/2$ and the parameter decreases with time through a quintessence region. A remark is in order. When $\epsilon_1 > 1$ the acceleration does not take place. However, it is clear that if $(3+\alpha) \sim \mathcal{O}(1)$ the acceleration phase with $\epsilon_1 \ll 1$ (namely, $H$ almost a constant) is immediately reached. For this reason we still indicate with $\mathcal{N}$ the total amount of inflation. On the other hand, for $\alpha < -3$, the $\epsilon_1$ parameter is negligible at the beginning of inflation and increases with the time.

The power spectrum of scalar perturbations is given by [48],

$$
\mathcal{P} = \frac{1}{8\pi^2\epsilon_1}\left(\frac{H^2}{M_{Pl}^2}\right)_{k=aH} \frac{2^{2\nu-1}}{\pi}|\Gamma(\nu)|^2, \quad \nu = \left|\alpha + \frac{3}{2}\right|.
\tag{85}
$$

As a check, note that when $\alpha = -3$ we recover the result of slow-roll inflation. Furthermore, the spectral index $n_s$ is related to $\alpha$ as

$$\alpha = \frac{1 - n_s}{2}, \frac{n_s - 7}{2}, \tag{86}$$

such that in order to obtain $n_s = 0.96$ with $\alpha > -3$ (the model in Equation (76)) we must fix $\alpha = 0.02$, while with $\alpha < -3$ (the model in Equation (77)) we must require $\alpha = -3.02$, namely we are near the slow-roll approximation region. Finally, the tensor-to-scalar power spectra ratio is still related to $\epsilon_1$ as

$$r = 16\epsilon_1, \tag{87}$$

where we remember that $\epsilon_1$ must be evaluated at the time when perturbations cross the horizon, at $N = 55 - 65$. In the case of $\alpha = 0.02$, in order to satisfy the Planck data with $r \lesssim 0.06$ we should require that $61.1 \lesssim \mathcal{N}$ if $\epsilon_1$ is evaluated at $N = 60$, namely the total time of inflation must exceed by at least one the number of *e*-folds of the perturbation horizon crossing. Moreover, for $\alpha = -3.02$ the tensor-to-scalar ratio is in agreement with the Planck data, leading to $r \simeq 0.03$ if $\epsilon_1$ is evaluated at $N = 60$.

Now, the TCC condition leads to

$$\epsilon_1 < \frac{10^9}{(8\pi)^2} e^{-2\mathcal{N}} \frac{2^{2\nu - 1}}{\pi} |\Gamma(\nu)|^2, \tag{88}$$

where $\epsilon_1$ is given by Equations (83) and (84) with $N = 60$.

Let us have a look for the viable model with $\alpha = -3.02$. The sufficient condition for the validity of the TCC can be found as in the slow-roll inflation scenario and is realized when $\mathcal{N}$ assumes the minimal value, namely $\mathcal{N} = 60$, when it is clearly violated. We remark that in this case we are near the slow-roll approximation region. In [48] it is argued that one can obtain $r \simeq 3 \times 10^{-3}$ (like in the Starobinsky inflation) by setting $\phi \sim M_{Pl}$ at the beginning of inflation. Here the result can be derived directly from Equations (84) and (87) by assuming $\mathcal{N} \simeq 60$. Thus, the model is affected by the trans-Planckian problem.

The situation is different for constant-roll inflation with $\alpha = 0.02$. In this case the $\epsilon_1$-parameter decreases with the *e*-folds and the sufficient condition for the validity of the TCC is realized when $\mathcal{N}$ is much larger than its minimal value for viable inflation, namely $\mathcal{N} \gg 61.1$. Specifically, we find that for

$$86.4 \lesssim \mathcal{N}, \tag{89}$$

the TCC condition in Equation (88) is satisfied. The results show that in the constant-roll inflationary scenario it is is possible to deal with viable inflation in agreement with the TCC provided that inflation starts much before the time when perturbations cross the Hubble horizon. In addition, in this case the tensor-to-scalar ratio $r$,

$$r < 3 \times 10^{-68}, \tag{90}$$

is extremely small and the model predicts a strong suppression of the amplitude of gravitational waves. However, at the beginning of inflation $\epsilon_1 = 1.51$ and is large enough to avoid a fine-tuning problem of the initial conditions, thanks to its peculiar behavior.

## 6. Conclusions

In this paper we revisited the Trans-Planckian Censorship Conjecture in different models of viable inflation. As already observed in [26] and, more recently, in [56], the TCC tightly constrains slow-roll scalar field inflation. Here, we first extended the result to different frameworks of slow-roll inflation. For scalar field theory and $f(R)$-gravity, we used a general approach that permits reconstruction of the models that lead to the power spectrum of scalar perturbations, spectral index and tensor-to-scalar spectra ratio in agreement with Planck data and where the TCC holds true. For $f(R, \phi)$-inflation, we

proposed the study of two viable models. In these cases, we found that although under certain conditions we can obtain viable inflation free of the trans-Planckian problem, we get a severe fine-tuning of initial conditions. Moreover, these models predict a strong suppression of the amplitude of primordial gravitational waves as a direct consequence of the TCC. In the second part of the paper, moving away from the slow-roll scenario, two examples of constant-roll scalar field inflation were analyzed. Here we found that in principle it is possible to deal with viable inflation avoiding both the trans-Planckian problem and fine-tuning problem of initial conditions, by asking that inflation starts much before the time when perturbations cross the Hubble horizon. We should stress that the result is related to a peculiar mechanism of inflation where the $\epsilon_1$ slow-roll parameter decreases with cosmological time and the de Sitter expansion is an attractor of the system, such that it is known that a transition phase at the end of inflation must be introduced.

We may conclude that, since in order to avoid the presence of fluctuations that trace back to quantities beyond the Planck scale in the classical power spectrum the TCC imposes severe constraints on the majority of the inflationary models, different mechanisms (as in slow-roll inflation) or different approaches (like the cosmological bounce) are the natural implications of the conjecture itself.

A last remark is in order. As briefly considered in the introduction, it should again be emphasized that in the TCC, as well as in the consideration of the trans-Planckian problem, the investigation of the generation of metrics and fields is restricted only to the scenario where metric fluctuations become large and quasi-classical, which may be thought of as a deficiency of the conjecture. Specifically, the important case of particle creation when metric and field fluctuations remain quantum but show themselves in the form of ultra-high energy particles is neglected. In this instance, as shown in [18], any deviation of the quantum state of trans-Planckian modes from the adiabatic vacuum one would result in the appearance of super-high energy particles in any expanding universe and at any time, including the present time.

**Author Contributions:** Both authors contributed equally to this work. Both authors have read and agreed to the published version of the manuscript.

**Funding:** This research received no external funding.

**Institutional Review Board Statement:** Not applicable.

**Informed Consent Statement:** Not applicable.

**Data Availability Statement:** Not applicable.

**Acknowledgments:** The authors wish to thank D. Grasso and G. Marozzi for useful discussions.

**Conflicts of Interest:** The authors declare no conflict of interest.

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
