# Peer review of "The Trans-Planckian Censorship Conjecture in Different Frameworks of Viable Inflation"

_universe, doi:10.3390/universe7040095_

Round 1
Reviewer 1 Report
The paper seems interesting and worth to publish. However, some changes and improvements have to be done.
First, I recommend to read the text attentively and correct English. For instance, terms containing “Plank” should be corrected throughout the paper.
Second, I suggest to add to conventions (in line 40) \hbar=1. Otherwise, with the standard definition of the Planck mass, the authors should add \hbar^{-1} in the gravity part of actions (3), (33), (54).
Next, the definition of V_\phi has to be added to eq. (6). Eq. (12) has to be corrected. P_\phi after eq. (15) has to be replaced with {\cal P}_\phi.
Further, I propose to add Appendix with details of the correspondence between eqs. (36), (37) and eqs. (8) and (10); the same concerns eqs. (55), (62) and eqs. (7), (10). Since actions (3), (33) and (54) are apparently related to each other, the same is expected for the mentioned characteristics.
Finally, at least two papers, by Linde (PLB 108 1982, 389) and by Albrecht&Steinhardt (PRL 48 1982 1220), have to be added to the reference list.
Author Response
We would like to thank the referee for his useful comments and suggestions. We embrace the referee suggestions and have modified the text according to his comments. Please find below our detailed reply to the referee suggestions:
First, I recommend to read the text attentively and correct English. For instance, terms containing “Plank” should be corrected throughout the paper.
We improved the english throughout the paper (including “Planck”!)
Second, I suggest to add to conventions (in line 40) \hbar=1. Otherwise, with the standard definition of the Planck mass, the authors should add \hbar^{-1} in the gravity part of actions (3), (33), (54).
We completed the sentence at the end of introduction as
“In our convention, the speed of light and the reduced Planck constant are c = \hbar=1.”
Next, the definition of V_\phi has to be added to eq. (6). Eq. (12) has to be corrected. P_\phi after eq. (15) has to be replaced with {\cal P}_\phi.
In Eq. (6), we replaced V_\phi with d V(\phi)/d\phi according with Eq. (12). We replaced P_\phi with {\cal P} after Eq (9) and Eq (15).
Further, I propose to add Appendix with details of the correspondence between eqs. (36), (37) and eqs. (8) and (10); the same concerns eqs. (55), (62) and eqs. (7), (10). Since actions (3), (33) and (54) are apparently related to each other, the same is expected for the mentioned characteristics.
We checked the consistency of our formalism as suggested by the referee. Therefore, under Eqs (59)-(60) with the power spectrum of f(R, phi)-gravity we showed that Eq (8) with the power spectrum of scalar field inflation and Eq (36) with the power spectrum of f(R)-gravity can be derived as special cases.
In an analogous way, under Eq (62), we showed that the spectral index and the tensor-to-scalar spectra ratio of scalar field and f(R) inflation can be derived as special cases of f(R, phi)-theory.
Finally, at least two papers, by Linde (PLB 108 1982, 389) and by Albrecht&Steinhardt (PRL 48 1982 1220), have to be added to the reference list.
We cited the suggested references at the beginning of introduction.
Reviewer 2 Report
The paper discusses the Trans-Planckian Censorship Conjecture in viable models of inflation in gravity theories beyond General Relativity.
Specifically, after reviewing the TCC the authors discuss slow-roll inflation models in f(R) and f(R,φ) theories, as well as constant-roll scalar field inflation. The paper is well written and the results are very interesting.
However, the f(R) section seems a bit incomplete. The authors find a "simple analytic" solution that reduces to the Starobinsky inflation, which is conformally equivalent to the model already presented in the previous section. Together with this, they also discuss the limiting case when f(R) reduces to the Einstein-Hilbert action.
It is not so clear to me why "f(R)-gravity models compatible with the TCC seem to bring to a strong suppression of the amplitude of the primordial gravitational waves and to a severe fine tuning of initial conditions" which is the concluding statement of Sec. 3. What would be nice to see is, either an f(R) inflation model where TCC is satisfied (in order to strengthen the afore-mentioned statement), or (even better) what are the necessary requirements for f(R) inflation models in order for TCC to be satisfied.
Other than that, for self-consistency of the paper, it would be nice to explain what α and β are after Eq. 17, as well as ε_3 after Eq. 35.
Author Response
We would like to thank the referee for his useful comments and suggestions. We embrace the referee suggestions and have modified the text according to his comments. Please find below our detailed reply to the referee suggestions
It is not so clear to me why "f(R)-gravity models compatible with the TCC seem to bring to a strong suppression of the amplitude of the primordial gravitational waves and to a severe fine tuning of initial conditions" which is the concluding statement of Sec. 3. What would be nice to see is, either an f(R) inflation model where TCC is satisfied (in order to strengthen the afore-mentioned statement), or (even better) what are the necessary requirements for f(R) inflation models in order for TCC to be satisfied.
In order to clarify this point, and due to the fact that, as observed by the referee, our analysis is not exhaustive, at the end of Chapter 3 we introduced the following paragraph:
“Despite the fact that our analysis is not exhaustive of the wide variety of f(R)-models for inflation, we can draw some conclusions. Since viable f(R)-gravity reduces to Einstein gravity at small curvature, it is clear that the TCC condition given in Eq. (39) introduces an upper bound on the effective EoS parameter as (1 +ωeff)< ∼10^23, such that the tensor-to-scalar spectra ratio in (37) results to be at most r∼10^44. This result is independent of the ansatz on ω_eff and is in agreement with (52)–(53). Thus, the suppression of the amplitude of the primordial gravitational waves seems to be a general feature of viable sow-roll f(R)-gravity compatible withTCC and since we have assumed as a minimal requirement that the total amount of inflation coincides with the e-folds at the perturbation horizon crossing, we get the fine-tuning problem of initial conditions.”
Other than that, for self-consistency of the paper, it would be nice to explain what α and β are after Eq. 17, as well as ε_3 after Eq. 35.
After Eq 17 we specified:
“where α and β are positive numbers. ”
After Eq 35 we added a footnote:
“In the next section we will consider the more general framework of f(R,φ)-gravity which includes f(R)-gravity as a special case. Thus, epsilon 1, epsilon 3 and epsilon 4 are labelled according with the corresponding slow-roll parameters in f(R,φ)-gravity”
and the following paragraph:
“We note that at the first order approximation the epsilon3 slow-roll parameter coincides with the opposite value of the epsilon1 slow-roll parameter and in the following expressions for the power spectrum and the spectral index we will pose epsilon3= −epsilon1 =dot H/H^2. However, the tensor-to-scalar spectra ratio must be evaluated at the second leading order of epsiloin1+epsilon3=( dot H/H2)epsilon4, what implicitly defines epsilon3.”
Reviewer 3 Report
In the reviewing paper, basing on Ref. [21], the authors compare a number of inflationary models with the so called Trans-Planckian Censorship Conjecture (TCC) with the result that these models do not satisfy it. This result has already been known for inflationary models based on minimally coupled scalar fields and f(R) gravity. However, some results in Sec. 4 and 5, where inflationary models based on f(R,\phi) gravity and constant-roll models are considered, are new. Thus, the paper contains some novel material.
Since the TCC has been announced as a conjecture, not as a theorem, I shall not criticize the authors for accepting it as it is. However, the well known deficiency of this conjecture, as well as of the consideration of the trans-Planckian problem in Refs. [9-11], is that the investigation of generation of metric and fields fluctuations in cosmology is restricted there only to the case when metric fluctuations become large and quasi-classical, so that the very important case of particle creation (including creation of gravitons) in cosmology when metric and field fluctuations remain quantum but show themselves in the form of ultra-high energy particles is neglected. As shown, in particular, in the papers by Starobinsky and Tkachev cited below, in this case any deviation of the quantum state of trans-Planckian modes from the adiabatic vacuum one would result in the appearance of super-high energy particles in any expanding universe, not necessarily expanding with acceleration, and at any time including the present time. I suppose that the authors have to acknowledge this deficiency of TCC, at least.
Follows is a number of remarks regarding citation.
1. First, regarding the citation of pioneer papers on a quasi-de Sitter (later dubbed inflationary) stage in the early Universe. The first observational test of this stage (the scale-free spectrum of primordial gravitational waves) was calculated in Ref. [8] before Refs. [1-3], and the first full model of it was developed in Ref. [36] which remains viable by now, in contrast to models later proposed in Refs, [1-3] and many other papers. Thus, I suppose that Refs. [8,36] have to be cited with Refs. [1-3] in the beginning of the paper in the chronological order. In fact, the inflationary model considered in Ref. [7] was the Starobinsky model [36].
2. Second, regarding Refs. [9-13]. All these pioneer inflationary models, as well as other existing viable inflationary ones like the Higgs inflation, are based on the minimal assumptions that the Lorentz invariance is not broken and additional spatial dimensions do not show themselves up to energies \sim 10^{14} Gev typical for these models for the number of e-folds N\sim 55 from the end of inflation. But then practically independently with Refs. [9-10], it was shown in A. A. Starobinsky, Robustness of the inflationary perturbation spectrum to trans-Planckian physics, JETP Lett. 73, 371 (2001), astro-ph/0104043, and later in A. A. Starobinsky and I. I. Tkachev, Trans-Planckian particle creation in cosmology and ultra-high energy cosmic rays, JETP Lett. 76, 235 (2002), astro-ph/020757, that there is no trans-Planckian problem for inflation under these assumptions. The absence of citation of these papers strongly misleads a reader. Of course, the situation can be different in string theories in the regime when the local Lorentz invariance is broken at some energy, but the value of this energy is still unknown.
3. Third, independently from Ref. [45], constant-roll solutions in f(R) gravity were studied in H. Motohashi and A. A. Starobinsky, f (R) constant-roll inflation, Eur. Phys. J. C 77, 538 (2017), arXiv:1704.08188 using a different definition of the constant-roll behaviour that gave a possibility to find explicit expressions for the function f(R) admitting exact constant-roll solutions. A more general case of constant-roll solutions in scalar-tensor gravity was more recently considered in H. Motohashi and A. A. Starobinsky, Constant-roll inflation in scalar-tensor gravity, JCAP 1911, 025 (2019), arXiv:1909.10833.
Thus, I suppose that this manuscript needs revision, after which it can be reconsidered for publication.
Author Response
We would like to thank the referee for his useful comments and suggestions. We embrace the referee suggestions and have modified the text according to his comments. Please find below our detailed reply to the referee suggestions
Since the TCC has been announced as a conjecture, not as a theorem, I shall not criticize the authors for accepting it as it is. However, the well known deficiency of this conjecture, as well as of the consideration of the trans-Planckian problem in Refs. [9-11], is that the investigation of generation of metric and fields fluctuations in cosmology is restricted there only to the case when metric fluctuations become large and quasi-classical, so that the very important case of particle creation (including creation of gravitons) in cosmology when metric and field fluctuations remain quantum but show themselves in the form of ultra-high energy particles is neglected. As shown, in particular, in the papers by Starobinsky and Tkachev cited below, in this case any deviation of the quantum state of trans-Planckian modes from the adiabatic vacuum one would result in the appearance of super-high energy particles in any expanding universe, not necessarily expanding with acceleration, and at any time including the present time. I suppose that the authors have to acknowledge this deficiency of TCC, at least
The limitations of the TCC have been acknowledged briefly in the introduction and later in the Conclusions, in more detail. The specifics are listed below.
Follows is a number of remarks regarding citation.
First, regarding the citation of pioneer papers on a quasi-de Sitter (later dubbed inflationary) stage in the early Universe. The first observational test of this stage (the scale-free spectrum of primordial gravitational waves) was calculated in Ref. [8] before Refs. [1-3], and the first full model of it was developed in Ref. [36] which remains viable by now, in contrast to models later proposed in Refs, [1-3] and many other papers. Thus, I suppose that Refs. [8,36] have to be cited with Refs. [1-3] in the beginning of the paper in the chronological order. In fact, the inflationary model considered in Ref. [7] was the Starobinsky model [36].
We rearranged the citations chronologically as suggested by the Referee.
Second, regarding Refs. [9-13]. All these pioneer inflationary models, as well as other existing viable inflationary ones like the Higgs inflation, are based on the minimal assumptions that the Lorentz invariance is not broken and additional spatial dimensions do not show themselves up to energies \sim 10^{14} Gev typical for these models for the number of e-folds N\sim 55 from the end of inflation. But then practically independently with Refs. [9-10], it was shown in A. A. Starobinsky, Robustness of the inflationary perturbation spectrum to trans-Planckian physics, JETP Lett. 73, 371 (2001), astro-ph/0104043, and later in A. A. Starobinsky and I. I. Tkachev, Trans-Planckian particle creation in cosmology and ultra-high energy cosmic rays, JETP Lett. 76, 235 (2002), astro-ph/020757, that there is no trans-Planckian problem for inflation under these assumptions. The absence of citation of these papers strongly misleads a reader. Of course, the situation can be different in string theories in the regime when the local Lorentz invariance is broken at some energy, but the value of this energy is still unknown.
We thank the Referee for the important remark. In this respect, we introduced the suggested papers as references in [17]-[18] in the introduction, and at the end of the conclusions we added the following paragraph taking into account the Referee's comments:
"A last remark is in order. As briefly considered in the introduction, it should be again emphasized that in the TCC, as well as in the consideration of the trans-Planckian problem,the investigation of generation of metric and fields is restricted only to the scenario where metric fluctuations become large and quasi-classical, which may be thought of as a deficiency of the conjecture. Specifically, the important case of particle creation when metric and field fluctuations remain quantum but show themselves in the form of ultra-high energy particles is neglected. In this instance, as shown in Ref. [18], any deviation of the quantum state of trans-Planckian modes from the adiabatic vacuum one would result in the appearance of super-high energy particles in any expanding universe and at any time, including the present time."
Third, independently from Ref. [45], constant-roll solutions in f(R) gravity were studied in H. Motohashi and A. A. Starobinsky, f (R) constant-roll inflation, Eur. Phys. J. C 77, 538 (2017), arXiv:1704.08188 using a different definition of the constant-roll behaviour that gave a possibility to find explicit expressions for the function f(R) admitting exact constant-roll solutions. A more general case of constant-roll solutions in scalar-tensor gravity was more recently considered in H. Motohashi and A. A. Starobinsky, Constant-roll inflation in scalar-tensor gravity, JCAP 1911, 025 (2019), arXiv:1909.10833.
At the beginning of Chapter 5 we cited the suggested references about constant roll inflation in f(R)-gravity as and in scalar tensor gravity as [51] and [52].
Round 2
Reviewer 3 Report
I am satisfied by the changes made by the authors and suppose that now this paper can be accepted for publication in the Universe journal.